# Identification and Validation of *Toxoplasma gondii* Mitoribosomal Large Subunit Components

**DOI:** 10.3390/microorganisms10050863

**Published:** 2022-04-21

**Authors:** Shikha Shikha, Mariana Ferreira Silva, Lilach Sheiner

**Affiliations:** Wellcome Centre for Integrative Parasitology, University of Glasgow, 120 University Place, Glasgow G12 8TA, UK; shikha.shikha@glasgow.ac.uk (S.S.); mariana.ferreirasilva@glasgow.ac.uk (M.F.S.)

**Keywords:** *Toxoplasma*, mitochondrion, ribosome, mitoribosome, apicomplexa, parasite, LSU

## Abstract

Mitochondrial ribosomes are fundamental to mitochondrial function, and thus survival, of nearly all eukaryotes. Despite their common ancestry, mitoribosomes have evolved divergent features in different eukaryotic lineages. In apicomplexans, the mitochondrial rRNA is extremely fragmented raising questions about its evolution, protein composition and structure. Apicomplexan mitochondrial translation and the mitoribosomes are essential in all parasites and life stages studied, highlighting mitoribosomes as a promising target for drugs. Still, the apicomplexan mitoribosome is understudied, with one of the obstacles being that its composition is unknown. Here, to facilitate the study of apicomplexan mitoribosomes, we identified and validated components of the mitoribosomal large subunit in the model apicomplexan *Toxoplasma gondii*.

## 1. Introduction

Apicomplexans are unicellular eukaryotic parasites that cause diseases with global impact on human and animal health. The phylum includes *Toxoplasma gondii,* which causes toxoplasmosis, a life-threatening disease of immuno-compromised people, and *Plasmodium* spp., causing malaria. Treatments for apicomplexan diseases are suboptimal, with side effects, cost and drug-resistance all posing treatment and eradication challenges. The parasite mitochondrial ribosomes (mitoribosomes) are promising targets for drug discovery because they are essential [1,2,3], because they are likely divergent from hosts, susceptible to specific inhibition [4,5], and because they offer opportunities for antibiotic repurposing [6]. However, the apicomplexan mitoribosome is understudied, with one of the main challenges being that its protein composition remains unknown. 

In addition to its potential as a drug target, an understanding of the apicomplexan mitoribosome composition and structure will address current questions in eukaryotic cell biology. Despite their common prokaryotic ancestry [7], mitoribosomes from unicellular eukaryotes present highly divergent features. Over the past decade, thanks to advances in cryo-electron microscopy, mitoribosomes of model organisms from different clades of the eukaryotic tree have been studied in detail, including representatives of the Opisthokonta (mammals [8,9] and yeast [10,11]), Archaeplastida (plants [12,13] and green algae [14]) and Excavata (Trypanosomatida [15,16]) clades. These studies revealed highly divergent protein content, RNA:protein ratio, size, and structural elements in mitoribosomes from different clades. However, there are so far no structural studies of mitoribosomes from a representative of the Chromalveolata group, which contains apicomplexa [17]. Moreover, the apicomplexan mitochondrial ribosomal RNA (rRNA) is transcribed in fragments, e.g., 27–34 fragments as shown in different *Plasmodium* spp. [18] and as can be predicted from the *Toxoplasma* mitochondrial genome sequence [19]. Mitoribosomal RNA fragmentation is seen also in the green algae *Chlamydomonas* (13 fragments), where novel *Chlamydomonas*-specific mitoribosomal proteins stabilize the rRNA in the large subunit (LSU) [14]. It is an intriguing question whether this is a common solution adopted in divergent lineages. Finally, the apicomplexan rRNA fragments, when assembled, are predicted to form the smallest rRNA molecules yet described [17,18,19,20,21]. The functional implications of small rRNA fragments are an intriguing avenue for future research. 

Thus, the apicomplexan mitoribosome poses exciting questions with impact on anti-parasitic drug discovery and on the broader field of cell biology. The first steps in addressing these questions are to identify a set of validated mitoribosomal components and to establish assays for mitoribosomal assembly and function in a tractable model organism. In a previous study we have provided validation for the essentiality, mitoribosome association and role in mitochondrial translation of a component of the small mitoribosome subunit (SSU) in *Toxoplasma*. Here we focused on validating *Toxoplasma* mitoribosome LSU components and provide further proof of concept for the assays established previously. Finally, we expand the list of *Toxoplasma* mitoribosomal protein candidates. 

## 2. Materials and Methods

### 2.1. In Silico Searches

Entries reported by Desai et al. [10], Amunts et al. [11], Waltz et al. [12], and Ramrath et al. [15] were used to obtain protein sequences from the yeast genome database https://www.yeastgenome.org/, uniprot, https://www.arabidopsis.org/index.jsp and from the *Trypanosome* genome database https://tritrypdb.org/tritrypdb/app. BLAST search and sequences homology/structure prediction were performed using tblastn (https://toxodb.org/toxo/app) and HHpred [22], respectively, and hits with e-value below 1 were recorded as potential homologs. All databases were accessed on 30 December 2021 for this analysis.

### 2.2. Parasite Culture 

All *T. gondii* lines were maintained in Human Foreskin Fibroblast (HFFs) in complemented DMEM media with 10% FBS, 1% Penicillin/Streptomycin antibiotics and 2% L-glutamine. All lines were grown at 37 °C and 5% CO_2_. 

### 2.3. DNA Cloning and Parasite Genetic Manipulation 

For CRISPR guided promoter replacement and endogenous tagging, the ChopChoP tool (http://chopchop.cbu.uib.no/, 18 January 2022) was used to identify gRNAs found around the ATG or STOP codon region of each gene. Each gRNA (Appendix A) was cloned into a vector containing the U6 promoter and expressing CAS9-GFP (Tub-Cas9-YFP-pU6-ccdB-tracrRNA) [23] using the BsaI restriction site. A DNA midi-prep (Qiagen, Manchester, UK) kit was used to purify the final plasmid, per the manufacturer’s protocol. Next, the pDT7S4myc plasmid was used as a template for amplification of DHFR selectable cassette and anhydrotetracycline (ATc) repressible promoter by PCR [24]. For tagging, the p3HA.LIC.CATΔpac plasmid was used as a template for amplification of CAT selection cassette and triple HA epitope by PCR [24]. Freshly egressed tachyzoites of TATiΔ*ku80* line were transfected with each corresponding gRNA/CAS9 vector and the PCR product mixture [24] by electroporation. Cassette integration was then selected with pyrimethamine or chloramphenicol. Serial dilution of positive pools was used to isolate clones and confirmed by PCR analysis (primers in Appendix A).

### 2.4. Immunofluorescent Assay

The HFFs on glass coverslips were infected with parasites and fixed with 4% paraformaldehyde after one day. Permeabilization and blocking was in 2% bovine serum albumin 0.2% triton X-100 in 1× PBS 20 min at room temperature (RT) before incubation with the following primary antibodies: anti-HA (Merck, Gillingham, UK) diluted 1:1000, rabbit anti-Tom40 [25] diluted 1:1000. This was followed by secondary antibodies (Alexa Fluor Goat anti-Rat 488 (Invitrogen #A-11006, Paisley, UK) diluted 1:1000, and Alexa Fluor Goat anti-Rabbit 594 (Invitrogen #A-11012, Paisley, UK) diluted 1:1000). Washes were in the same buffer as blocking. Fluoromount-G containing DAPI (Southern Biotech, Birmingham, AL, USA, 0100–20) was used for mounting. 

Images were acquired via a Delta Vision Core microscope (Applied Precision, Rača, Slovakia) with a magnification of 100×. The SoftWoRx and FIJI software were used to process and deconvolve images.

### 2.5. Growth Assay

The HFFs in 6-well plates were infected with *Toxoplasma* and grown in the presence or in the absence of ATc 0.5 µM for nine days. Methanol 100% was used to fix the infected HFF monolayer for 30 min at RT and crystal violet dye was applied to stain for 2 h at RT. 1× PBS was used for washes after each step. Plaques were imaged with a phone camera. 

### 2.6. Native and Denaturing Gels, in Gel Activity Assay and Western Blot

Separation of tachyzoites proteins were performed via SDS-PAGE where the parasite suspension was collected, centrifuged, and the remained pellet resuspended in 1 × NuPAGE LDS loading dye (Invitrogen, Paisley, UK) supplemented with 5% *v*/*v* beta-mercaptoethanol and boiled at 95 °C for 5 min. Semi-dry transfer in Towbin buffer (0.025 M Tris 0.192 M Glycine 10% Methanol) was incubated for 60 min at 190 mAmps onto nitrocellulose membrane (0.45 μm Protra™, Merck, Gillingham, UK). The blots were blocked in milk and labelled with the relevant antibodies: primary rat anti-HA (1:500, Sigma, Merck, Gillingham, UK), rabbit anti-Tom40 [25] labelled with secondary horseradish peroxidase (HRP) conjugated antibodies (Promega for mouse and rabbit, and Abcam for rat) (dilution 1:10,000) and developed with Pierce ECL Western Blotting Substrate (Thermo Scientific, Paisley, UK) or coupled with secondary fluorescent antibodies IRDye® 800CW (1:10,000, LIC-COR, Lincoln, NE, USA) and visualized with Odyssey LCX imaging system. 

For blue-native (BN) PAGE, the parasite pellet was suspended in BN-solubilisation buffer (750 mM aminocaproic acid, 0.5 mM EDTA, 50 mM Bis-Tris–HCl pH 7.0, 1% digitonin or DDM), incubated for 15 min on ice and then centrifuged at 16,000× *g* at 4 °C for 30 min. The resulting supernatant had sample buffer added containing Coomassie G250 (NativePAGE) to a final concentration of 1% detergent and 0.0625% Coomassie G250. An equivalent of 1–2 × 10^7^ parasites was loaded per each lane and separated on a Native PAGE 4–16% or 3–12% Bis-Tris gel. For the molecular marker, NativeMark^TM^ was used. 

For immunoblotting, the transfer of proteins was onto a PVDF membrane (0.45 μm, Hybond, Merck, Gillingham, EN, UK) via wet transfer in Towbin buffer. The transfer was for 60 min at 100 V. Immuno-labelling and detection was as described above. 

The Clear-native (CN) PAGE along with complex IV activity assay was performed according to [3]. Briefly, parasite pellets were suspended in CN-solubilisation buffer (50 mM NaCl, 2 mM 6-aminohexanoic acid, 50 mM Imidazole, 2% (*w*/*v*) n-dodecylmaltoside, 1 mM EDTA–HCl pH 7.0), following 10 min incubation on ice and then centrifugation at 16,000× *g* at 4 °C for 15 min. The resulting supernatant was combined with glycerol and ponceau S (with final concentrations of 6.25% and 0.125%, respectively). An equivalent of 2 × 10^7^ parasites per lane was separated on a Native PAGE 4–16% Bis-Tris gel along with the molecular weight marker, NativeMark. The resulting gel was incubated with Complex IV oxidation buffer (1 mg mL^−1^ cytochrome *c*, 50 mM KH_2_PO_4_, pH 7.2, 0.1% (*w*/*v*) 3,3′-diaminobenzidine tetrahydrochloride).

### 2.7. Immuno-Precipitation

About 1 × 10^8^ freshly egressed parasites were collected via 1500× *g* spin for 15 min at 4 °C followed by lysis on ice for 15 min with lysis buffer (1× TBS, 1% DDM (Thermo Fisher, Paisley, UK), 1 mM DTT (Sigma) and 0.4 U/μL of RNaseOUT™ Recombinant Ribonuclease Inhibitor (Thermo Fisher, Paisley, SC, UK). The lysates were spun at 16,000× *g* for 10 min at 4 °C. 50 μL of washed Pierce® Anti-HA Agarose beads (Thermo scientific, Paisley, UK) were used to pull-down the supernatant according to the manufacturer’s instruction. 

### 2.8. RTqPCR

For each experiment, the parasites were cultured in triplicate in absence or presence of ATc for 2 days. Freshly egressed tachyzoites were collected for RNA extraction with RNeasykit (Qiagen, Manchester, UK) with the addition of a DNAseI step. Following the RNA-to-cDNA kit manufacture’s protocol, cDNA was made and qRT-PCR was set up using PowerSYBR green master mix (ThermoFisher, Paisley, UK) with 10 ng of cDNA as a template. Specific primers for mito-RNA, api-RNA and actin were used with annealing temperature of 52 °C and elongation time of 30 s. Twenty-five cycles were performed using 7500 Real Time PCR System (Applied Biosystems, Waltham, MA, USA). To calculate the relative expression of samples treated or not with ATC, the double Δ Ct method [26] was used with actin mRNA as an internal control. The software GraphPad Prism 9.2.0 was used for data plotting. Results from the different treatments were compared with an unpaired *t*-test.

## 3. Results

### 3.1. Identification of New Toxoplasma LSU Component Candidates

Previous studies generated an inventory of putative apicomplexan mitoribosomal proteins based on homology to other organisms [3,27]. However, a series of recent structural studies expanded the pool of confirmed mitoribosomal proteins from an array of divergent organisms. We thus decided to revise the inventory through BLAST searches for *Toxoplasma* homologs of the mitoribosomal proteins reported in the mitoribosomal structures of one representative of the ophistokont (yeast), plant (*Arabidopsis*) and Excavata (Trypanosomes) clades, via ToxoDB (https://toxodb.org/toxo/app, 18 January 2022) (Appendix A). We found that tBLASTn provided inconsistent results, whereby searching with orthologs of the same component from different organisms provides different *Toxoplasma* hits (for example, *Arabidopsis* bL21m hits TGGT1_202350, while *Trypanosoma* bL21m hits TGGT1_210360, and yeast bL21 had no hits) (red rubrics in Appendix A). We thus decided not to pursue hits from this search experimentally. 

Next, we aimed to use HHPRED (https://toolkit.tuebingen.mpg.de/tools/hhpred, 18 January 2022) as means to analyse *Toxoplasma* genes for their putative homology to known mitoribosomal proteins from the new structures. In order to focus this search on a small group of genes, and in an attempt to identify genes that are typically missed in other searches, we used the ToxoDB strategy function to select genes encoding hypothetical proteins, with no localization data available in the Localization of Organelle Proteins by Isotope Tagging (LOPIT) dataset [28]. To enhance the focus on mitochondrial proteins, we excluded proteins with a predicted signal peptide. As we expected mitoribosomal proteins to be essential for *Toxoplasma* tachyzoites in culture, we selected genes with an essentiality score that predicts high contribution to fitness (−2 to −6.89). This yielded 494 genes (Appendix A, strategy tab, shows the strategy schematics). Next, we aimed to focus on proteins that have potential orthologs in *Plasmodium falciparum*, in order to identify conserved apicomplexan components. An EupathDB orthology search identified 111 hits that answer this criterion (Appendix A, “111” tab). Finally, to continue our focus on new genes not studied before we selected the 42 genes whose *Plasmodium* ortholog encode “conserved protein, unknown function”, and those were used to run searches in HHPRED (Appendix A, “42” tab). A scheme summarizing the search pipeline is found in Appendix A. We found eight *Toxoplasma* homologs of known mitoribosomal proteins not identified previously. Of those, three encode SSU components (TGGT1_233170, TGGT1_248790, TGGT1_267340), two are not predicted to be mitochondrial using the MitoProt tool (https://ihg.gsf.de/ihg/mitoprot.html, 18 January 2022) (TGGT1_213570, TGGT1_240270), and one (TGGT1_207020) has a *Plasmodium* homolog (PF3D7_1204700) with a user comment in its PlasmoDB gene page that suggests a different function. We thus selected the two remaining LSU components homologs, TgbL35m (TGGT1_320005) and TgbL36m (TGGT1_222180). These are new *Toxoplasma* LSU components candidates not identified before, and the function of their homologs has not been studied in any of the other systems; therefore, we decided to test them experimentally. 

In addition to TgbL35m and TgbL36m, we previously identified TGGT1_226280 within an in silico screen for mitochondrial proteins [3]. In this previous study, we localized its gene product to the mitochondrion using expression of a tag minigene [3], and its role within the mitoribosome awaits validation. The encoded protein has homology to bL28m, and thus we name it here TgbL28m, and added it to our list of targets for experimental validation. Finally, a recent study of translating mitoribosomes suggested that uL24m is involved in the delivery of newly synthesized polypeptides to the mitochondrial inner membrane insertase Oxa1L [29]. Due to this central role, we selected the *Toxoplasma* homolog of TguL24m (TGGT1_216010 [3,22]) for validation. Overall, four predicted LSU component homologs were selected for experimental validation (Table 1).

### 3.2. Endogenous Tagging Confirms the Mitochondrial Localization of TguL24m and Provides Evidence for Its Association with the Toxoplasma Mitoribosome 

It was expected that mitoribosomal proteins would localize to the mitochondrion. To examine the localizations of our proteins of interest, we used endogenous gene tagging via single homologous recombination to introduce a triple-HA epitope tag fusion at the protein C-terminal as done before [30] (Appendix A). Correct integration of the tagging cassette was confirmed by PCR (Figure 1A and Appendix A). Signal colocalizing with the mitochondrial marker Tom40 [25] was seen for TguL24m confirming its mitochondrial localization (Figure 1B). For TgbL35m and TgbL28m, no signal was detectable by immunofluorescence or via western blot following native migration (Appendix A). This might be the result of low expression levels, and the observation that TgbL35m peptides are also not found in the dataset from the Toxoplasma LOPIT [28] supports this possibility for those two genes. We were unable to isolate lines with endogenously tagged TgbL36m.

Proteins of the *Toxoplasma* mitoribosome migrate in a high molecular weight band on a native protein gel (above the 1048 kDa marker band) [3]. We examined native gel migration for TguL24m via migration on a native PAGE followed by western blot, alongside the previously studied TgmS35-HA. As expected for components of the same complex, TguL24m migrated at a similar size to TgmS35-HA (Figure 1C). 

Finally, we have previously demonstrated that immunoprecipitation (IP) of the SSU mitoribosomal component TgmS35 enriches mitochondrially encoded SSU rRNA [3]. We examined this for TguL24m. As before, RNA was extracted from the IP elution (Appendix A), followed by reverse transcriptase reaction and PCR for the SSU rRNA or for the un-related cytosolic actin mRNA. A band of the SSU rRNA could be amplified while actin mRNA could not (Figure 1D). Collectively, these observations are in support of TguL24m being part of the mitoribosome. 

### 3.3. Depletion of Each of TgbL35m, TgbL36m and TgbL28m Results in Toxoplasma Growth Defect

For the three genes where a tag could not be detected preventing us from performing the analysis done for TguL24m, we turned to a genetic depletion approach. Mitochondrial translation is essential for *Toxoplasma* (tachyzoite) growth [3], thus it is reasonable to predict that genes encoding mitoribosomal components would each be essential. Fitness scores from a *Toxoplasma* genome wide CRISPR-based screen [31] support this prediction for the three LSU component homologs TgbL35m, TgbL36m and TgbL28m (Table 1). To provide validation for these predictions we constructed inducible knock-down lines for each gene by replacing the native promoter with an ATc repressible promoter in a parasite line with enhanced homologous recombination as described previously [24]. Each promoter replacement was performed via CRISPR/Cas9 as shown in (Figure 2A) and as described previously [3]. The correct integration of the new promoters was confirmed by PCR (Figure 2B), and ATc induced downregulation of each of the corresponding mRNAs was confirmed by RT-qPCR (Figure 2C). Plaque assays were used to examine parasite growth upon depletion of each one of the genes, showing growth defect in all cases (Figure 2C). These results indicate that the LSU component homologs TgbL35m, TgbL36m and TgbL28m are essential for Toxoplasma growth.

### 3.4. Depletion of Each of TgbL35m, TgbL36m and TgbL28m Results in a Mitochondrial Translation Defect 

It is expected that depletion of a mitoribosomal component necessary for assembly or function would result in a mitochondrial translation defect. In lieu of a reliable direct mitochondrial translation assay, we utilized an indirect assay, which we had established in our previous study of the SSU component TgmS35 [3]. In this assay we monitor the assembly and activity of the respiratory chain complex IV, which is directly dependent on the mitochondrial translation of its CoxI and CoxIII components, and compare it to the assembly of complex V (ATP synthase), which is not directly dependent on mitochondrial translation, as all of its subunits are nuclear encoded [26,32,33,34]. For each depletion line, lysates were collected from ATc treated and un-treated parasites, following 48, 72 or 96 h of incubation with ATc. Clear-native PAGE was performed, followed by an in-gel complex IV enzymatic assay, or by western blot with anti-ATP-synthase beta subunit, which is used to detect complex V. For all three lines, the ATc treated parasites showed reduced signal from the complex IV enzymatic assays compared to the un-treated control, while the signal from complex V remained unchanged (Figure 3A). As a control for complex V defect detection, we showed that upon depletion of the mitochondrial protein import component TgTom22 [25] the signal from complex V in our assay was fully reduced (Figure 3B). These observations support a role for components TgbL35m, TgbL36m and TgbL28m in mitochondrial translation and provide evidence for their mitoribosomal affiliation. 

### 3.5. A Genetic Interaction Occurs between Genes Encoding LSU Components in Toxoplasma 

We previously showed that depletion of one *Toxoplasma* SSU component (TguS15m) results in downregulation of another (TgmS35) [3]. We examined if this genetic interaction occurred also between LSU components. For this we performed RTqPCR on RNA extracted from the TgbL35m repressible line following 48h of incubation with ATc. We found that the mRNA for the genes encoding TgbL28m, TgbL36m, TguL24m and TgmS35 were downregulated under these conditions (Figure 4). These finding point to a genetic interaction between the herein studied LSU component homologs and provides further support for their affiliation in the same complex.

## 4. Discussion

Establishing markers and functional analysis assays to study the apicomplexan mitoribosome is an essential first step to address questions about its overall composition and its structure. Here we validate the essentiality, mitoribosome association, and role in mitochondrial translation of components of the mitoribosomal LSU in *Toxoplasma*. This work joins previous studies that validated and characterized the function of *Toxoplasma* SSU [3], *Plasmodium* LSU [2] and *Plasmodium* SSU [1] components, providing a complete repertoire of markers and cell-lines to study mitoribosomes in both these apicomplexan models. 

### 4.1. Identification of New Apicomplexan Mitoribosomal Proteins in This Study 

Our search via the HHPRED tool discovered mitoribosomal subunits that have been missed in previous searches (Appendix A). This is likely due to the accumulating structural data enabling the discovery of proteins that have structural homology and that cannot be found merely via sequence similarity. Finding new proteins may also have been thanks to the focus on a group of genes that is often overlooked, namely genes whose encoded proteins have no predicted functional domains or localization. While this provides some progress, there are still many known mitoribosomal proteins whose homologs are not yet found in *Toxoplasma*. Seeing that the HHPRED search with *Toxoplasma* sequences as queries was productive, future work could focus on this strategy, and screen groups of *Toxoplasma* genes selected based on different criteria, such as genes showing tight co-expression with known mitoribosomal proteins, or proteins that have evidence of being mitochondria localized. We took a first step at addressing the latter group and used the *Toxoplasma* LOPIT dataset as a starting point. Among proteins with high probability of being in the mitochondrial soluble fraction we selected hypothetical proteins which are likely essential according to their CRISPR screen phenotype score. We queried HHPRED with each of the resulting 58 proteins (Appendix A). We found six previously unknown potential mitoribosomal components, and two cases where a previously suggested homolog is challenged by the new data, which will be a starting point for future work.

### 4.2. Identification of New Apicomplexan Mitoribosomal Proteins with No Known Homologs 

Considering the growing number of examples of divergent eukaryotes where an evolutionary shift took place from rRNA towards proteins in the mitoribosomes (for example 1:6 in Trypanosomes [15]), and the proportion of organism specific components (at least 60 in Trypanosomes [15], and at least 11 in *Chlamydomonas* [14]), it is reasonable to hypothesize that the apicomplexan mitoribosome have many more components than those identifiable by sequence or structure based homology searches. One of the origins of this heterogeneity in mitoribosome composition is the evolution of gene families derived from a single ancestral gene [35]. For example, in plants, 16 mitoribosomal components are encoded by such small multigenic families [36]. In this context, a particular interest arises in the recently identified family of novel mitochondrial RNA binding proteins in apicomplexans [37]. It was proposed that these might play a role in ribosome assembly or stability, which is in line with the need emerging from the multiplicity of rRNA fragments, however this awaits to be studied. 

Other experimental methods could be used to discover new components not predicted through homology. Our ability to isolate *Toxoplasma* lines with endogenously tagged mitoribosome components, that are still integrated into a high molecular weight complex, raises the possibility of using proximity tagging methods to identify other complex components. In the absence of reliable structure prediction of how the validated components interact, it is hard to predict which proteins to tag and this would require trial and error. Alternatively, we observed co-depletion of several LSU mRNAs in response to depletion of another LSU component (Figure 4), which was also observed for two SSU components in our previous study [3]. These observations may reflect a pattern of genetic interaction between mitoribosome subunits, and thus might present a strategy to identify new mitoribosomal component candidates via analysis of changes in their expression in the various depletion mutants of known components.

### 4.3. Proof of Principle for Methods to Assay Apicomplexan Mitoribosome Assembly and Function

Some of the experiments performed here could now provide means for future functional studies. For example, our RT-PCR experiment performed from the TguL24m IP, provides evidence for an interaction between an SSU rRNA and an LSU protein; this is the first evidence for interactions between components of the SSU and LSU in the apicomplexan mitoribosomes. This interaction might be used as an indicator for monosome assembly in future studies, pending relevant controls. Likewise, the assay we had established previously as proxy for mitochondrial translation, received further proof of principle in the current study.

### 4.4. Validation That TguL24m, a Homolog of a Key Component of Nascent Chain Protection in Human Mitoribosome, Is Part of the Toxoplasma Mitoribosome

In a structural study of the human mitoribosome bound to the insertase Oxa1L, uL24m was found to play a pivotal role in limiting the folding of the nascent chain while still in the exit tunnel. This is likely important to allow the correct co-translational folding and insertion of the highly hydrophobic proteins encoded in the mitochondrial genome directly into the inner membrane. The main player in this structure is in fact, mL45, whose tail folds to create a constriction site that would limit helix formation of the nascent chain, and uL24m stabilizes this folding. The mitochondrial genome of *Toxoplasma* seems to have evolved to only encode highly hydrophobic proteins, as all soluble proteins encoded in other mitochondrial genomes have moved to the nucleus, sometimes at the cost of reduced hydrophobicity [34]. Thus, the role of protecting the nascent protein from premature folding might be even more central to the apicomplexan mitoribosome function. Structural analysis of the parasite mitoribosome is necessary to examine this question. Having validated the TguL24m homolog as a mitoribosomal component, this will hopefully be helpful to examine any insight from such structure. We were unable to identify an unambiguous candidate for a *Toxoplasma* mL45. 

## 5. Conclusions

With validated markers for apicomplexan mitoribosome SSU and LSU, and a series of functional assays, future efforts should be aimed to solve the structures of these divergent complexes, and to discover parasite specific components and features. We hope that the various new endogenously tagged lines and their corresponding mutants would provide means and controls for attempt to biochemically enrich *T. gondii* mitoribosome for proteomics and for structural analysis. A potential challenge for these efforts may well be maintaining the complex integrity through the necessary biochemical procedures. 

## Figures and Tables

**Figure 1 microorganisms-10-00863-f001:**
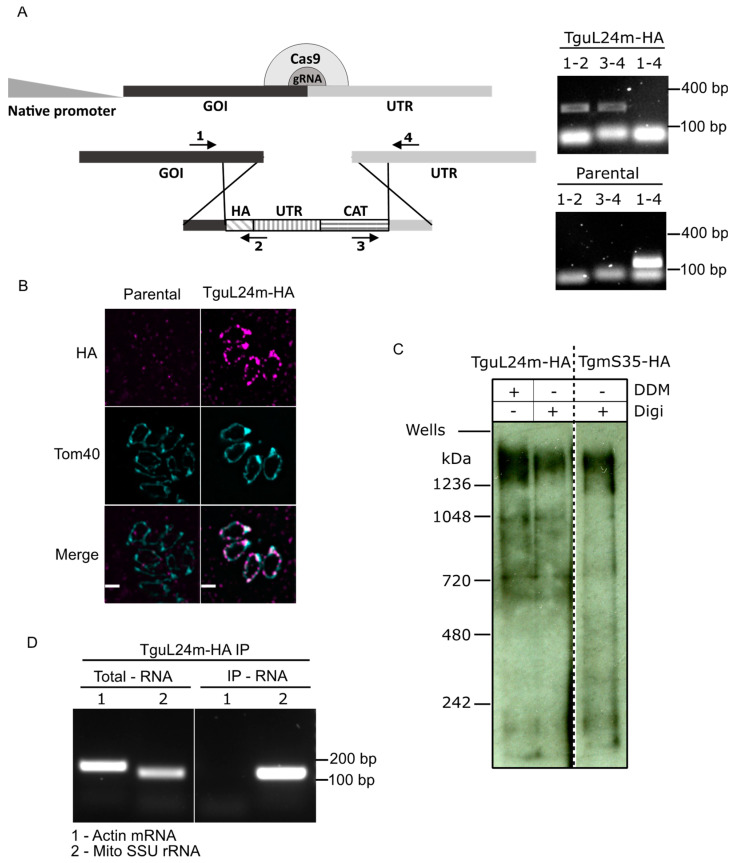
Evidence supporting the affiliation of a *Toxoplasma* uL24m homolog as a mitoribosomal component. (**A**) (Left) Scheme describing the genetic manipulation for endogenous HA tagging of Tg*uL24m*. (Right) Validation of the tag integration in the Tg*uL24m* locus via PCR analysis using primers 1, 2, 3, and 4, shown in the scheme on the left. (**B**) Immunofluorescence micrographs taken with the RH∆ku80/TATi parental cell line and TguL24m-HA tagged cell line, probed with anti-HA and anti-Tom40 antibody showing expression of the HA tag only in the tagged line and colocalization with the mitochondrial marker Tom40. Scale bar = 1 µm (**C**) Total cells from TguL24m-HA and TgmS35-HA cell lines treated with detergents Digitonin (Digi) and N-Dodecyl b-D-maltoside (DDM) as depicted, followed by separation on Blue-Native PAGE and immunoblotted to visualize HA tagged TguL24m and TgmS35 by probing with anti-HA antibody showing migration of TguL24m and TgmS35 at similar sizes regardless of detergent treatment. (**D**) RT-PCR experiment to detect mitochondrial small subunit rRNA (Mito SSU rRNA) performed on RNA sample extracted from total cells and eluate by immunoprecipitated TguL24m-HA using HA beads. RT-PCR for a cytosolic mRNA (Actin mRNA) was also performed on the same sample as negative control. The fact that TguL24m has a mitochondrial localization, high-molecular band migration and interaction with mitochondrial rRNA provide evidence for its affiliation with the mitoribosome.

**Figure 2 microorganisms-10-00863-f002:**
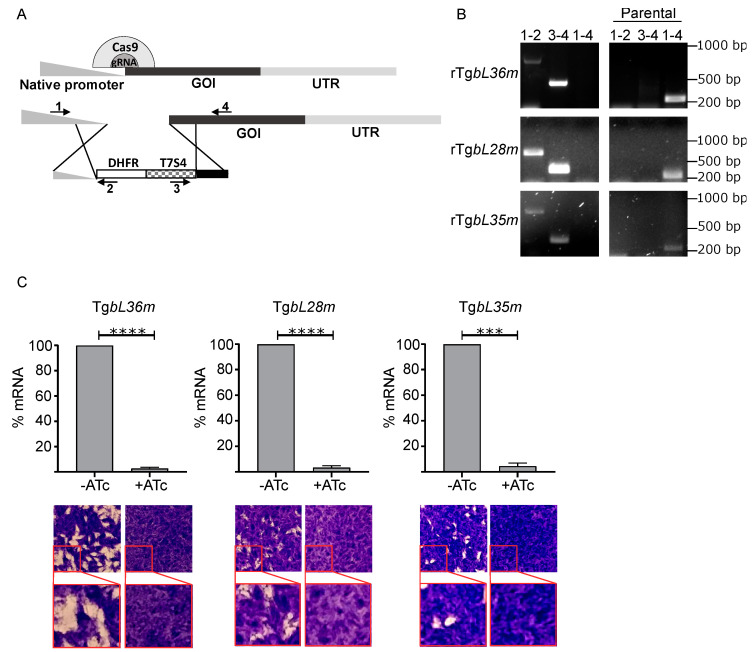
*Toxoplasma* LSU homologs are essential for parasite growth. (**A**) Scheme depicting the genetic manipulation leading to promoter replacement of genes mentioned in (**B**) and showing the location of the primers used in (**B**). (**B**) Validation of the repressible promoter integration in the Tg*bL36m*, Tg*bL28m* and Tg*bL35m* gene loci via PCR analysis using primers 1, 2, 3, and 4 shown in (**A**). (**C**) Bar graphs on the top show transcript levels of Tg*bL36m*, Tg*bL28m* and Tg*bL35m*, analysed by qRT-PCR, in absence (−ATc) or presence (+ATc) of Anhydrotetracycline (ATc) after 48 h (*n* = 3, bars represent mean ± SD, ***** p* < 0.0001, **** p* = 0.0009; analysed using unpaired *t*-test). Images on the bottom show plaque assays corresponding to the respective cell lines and respective condition above them. For plaque assays parasites were grown in the absence (−) or presence (+) of ATc for 9 days and host cell monolayer was stained with Crystal violet to visualize plaques, under each plaque assay images, there is an enlarged image of the region shown in the red squares to improve clarity. These findings validate that down regulation of either of the Tg*bL36m*, Tg*bL28m* and Tg*bL35m* genes result in growth defect.

**Figure 3 microorganisms-10-00863-f003:**
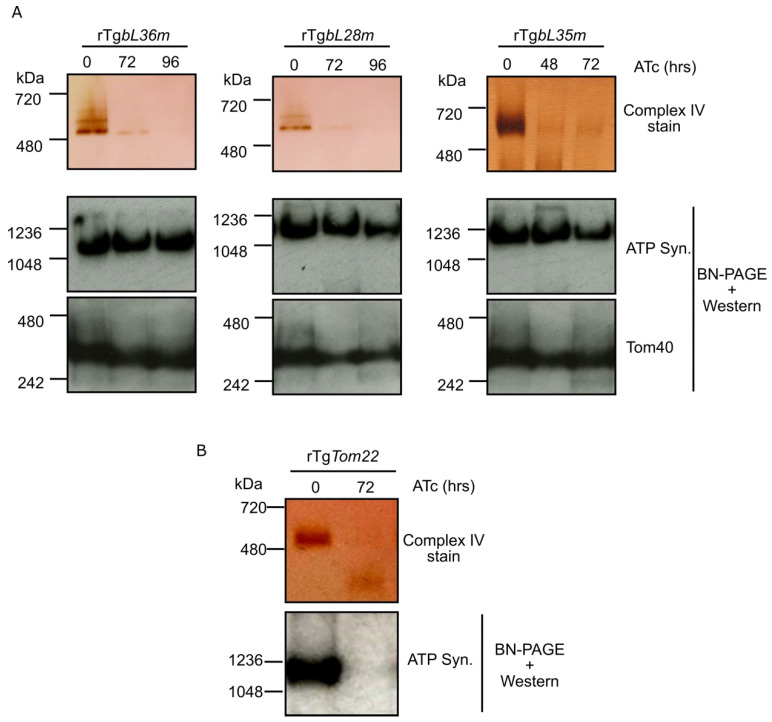
Depletion of Tg*bL36m*, Tg*bL28m* and Tg*bL35m* results in a defect in mitochondrial translation. (**A**) Whole cell lysate from Tg*bL36m*, Tg*bL28m* and Tg*bL35m* conditional knockdown cell lines, grown in the absence (0) or presence of ATc for 48, 72 or 96 h, separated by high resolution clear- native PAGE. Complex IV activity (top three panels) was visualized with cytochrome c: DAB staining. Complex V/ATP synthase assembly (middle three panels) was assayed by separating whole cell lysate from the respective cell lines on Blue-Native PAGE (BN-PAGE) followed by western blot using a commercial antibody raised against ATP synthase beta subunit (ATP Syn.), which is a part of complex V. Blots were also probed with anti-Tom40 antibody serving as loading control (lower three panels). (**B**) Complex IV and Complex V assembly assay as described in (**A**) performed on Tg*Tom22* knockdown cell line grown in the absence (0) or presence of ATc for 72 h. Loss of Complex V/ATP synthase assembly demonstrates its vulnerability when its components are depleted due to defect in the mitochondrial protein import providing a positive control that further validates the mitochondrial translation assay performed by comparing Complex IV and V assembly. Collectively, these finding point to a mitochondrial translation defect in the three lines.

**Figure 4 microorganisms-10-00863-f004:**
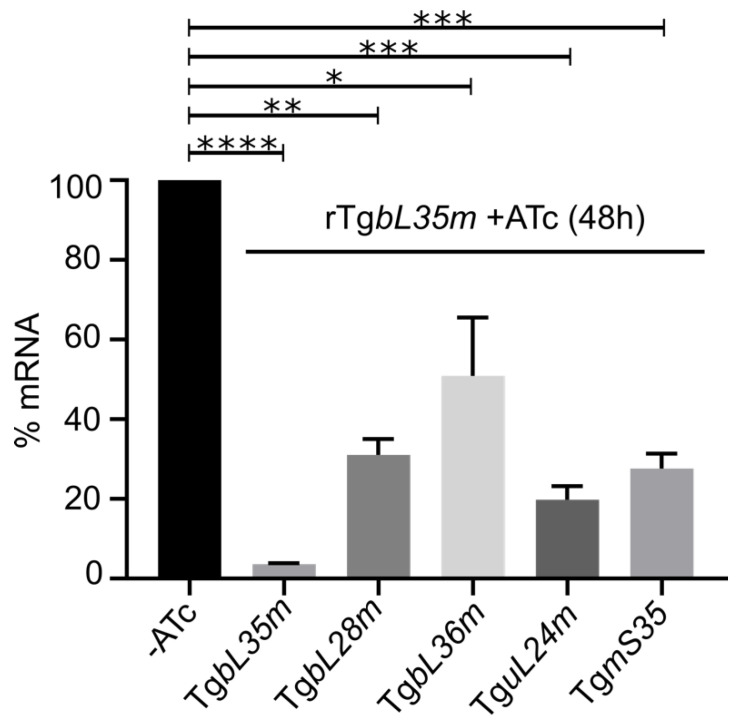
Depletion of Tg*bL35m* results in downregulation of other mitoribosomal subunit homologs. Bar graph depicting the transcript levels of Tg*bL35m*, Tg*bL28m*, Tg*bL36m*, Tg*uL24m* and Tg*mS35*, analysed by RT-qPCR, from RNA sample collected at 48h of ATc treatment of the Tg*bL35m* depletion cell line. Transcript levels for all the above genes in the absence of ATc are normalized to 100% and this is presented as one bar for -ATC (*n* = 3, bars represent mean ± SD, ***** p* < 0.0001, **** p* ≤ 0.0009, *** p* = 0.0011, ** p* = 0.0284; analysed using unpaired *t*-test). These finding show reduced expression of genes encoding mitoribosomal proteins in respond to depletion of another mitoribosomal protein.

**Table 1 microorganisms-10-00863-t001:** Summary of the four *Toxoplasma* LSU component homologs studied in this work.

Gene ID (Name)	Previous Publication/Homology	Mitoprot Score	CRISPR Screen Score	Plaque Defect?
TGME49_226280 (Tg*bL28m*)	[3,27]	0.9971	−4.91	Yes
TGME49_222180 (Tg*bL36m*)	homolog of Trypanosoma bL36m	0.7884	−3.96	Yes
TGME49_320005 (Tg*bL35m*)	homolog of Arabidopsis bL35m	0.9989	−4.99	Yes
TGME49_216010 (Tg*uL24m*)	homology of human uL24m	0.492	−3.66	--

## Data Availability

All geneIDs provided are found in EupathDB https://veupathdb.org/veupathdb/app, accessed on 18 January 2022. All hits from HHPRED searches are found in Uniprot https://www.uniprot.org/, accessed on 18 January 2022.

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
