# Peer review of "Identification and Validation of Toxoplasma gondii Mitoribosomal Large Subunit Components"

_microorganisms, 2022, doi:10.3390/microorganisms10050863_

Round 1

Reviewer 1 Report

The manuscript from Shikha et al. describes the characterisation of novel mitoribosomal proteins in Toxoplasma gondii. The authors start from an in silico search specifically designed to identify proteins that are part of the mitoribosome but might be missed by traditional homology searches, due to the evolutionary divergence of this complex. The combination of multiple parameters, including the predicted essentiality of the complex, conservation with Plasmodium and structural homology (HHPRED) to previously characterised components in different clades, lead to the identification of 2 candidates of the large subunit (LSU). The identified proteins were studied, together with 2 other previously identified putative LSU components, using genetic & biochemical approaches. The results provide evidence that supports the identification of these 4 proteins as members of the mitoribosome LSU ad their role in mitochondrial translation.

The work adds to our current knowledge of the Toxoplasma mitoribosome, a macromolecular complex that is essential for parasite life and a possible drug target and sets the stage for further work to define its composition. The experimental design is well though-out and clearly explained. The experimental data and discussion are also well presented and articulated. I think this work will be of interest to the Toxoplasma research community and to those interested in understanding mitoribosome divergence in the different eukaryotic clades.

I only have some minor comments and suggestions that I hope will be helpful to improve clarity and readability.

- While in situ tagging with HA of TgbL28m was not successful in this work, TgbL28m was experimentally localised to the mitochondrion in Lacombe et al. 2019 using minigene expression and a myc tag. I think it would be worth to underline this point, as in the current manuscript it doesn't come across very clearly. If the authors think it appropriate, discuss why in situ tagging might have failed and what are the possible cons of the minigene approach.

- The first question that comes to my mind reading this work is to ask if the tagged cell lines described here or in Lacombe et al. can be used to enrich for the mitoribosome and identify its more divergent components by mass spectrometry. As a follow-up, could proximity-dependent biotinylation or crosslinking approaches be of use in this case? I think the discussion would benefit from commenting on the pro & cons of these strategies in the characterisation of the Toxoplasma mitoribosome.

- Figure 1A. It would be helpful for the reader to know where the primers are binding without having to go to Figure S1. I think this could be done simply by modifying the legend to describe the primers, e.g.: Primer 1 GoI Fwd; primer 3 CAT forward, etc. Alternatively, panel A could be moved to Figure S1.

-Figure 4. If I understood correctly, all transcript levels for the mitoribosome components have been normalised to 100% in the -ATc conditions and this is shown as a single -ATc bar. I think this doesn't come across readily and the legend might need to be revised a bit to make this clearer to the reader.

- Figure S1B: the legend should be revised as IFA data is shown only for the two cell lines that did not give a clear signal and it is not very clear that F3 is the parental line.

- Line 381: I think it should say genes instead of genomes.

- There are some typos in lines 147, line 199, line 152 and lines 162-164 should be removed

Reviewer 2 Report

Shikha and colleagues presented a manuscript intitled “Identification and validation of Toxoplasma gondii mitoribosomal large subunit components”. This constitutes an original research article aiming to identify apicomplexan mitoribosomes, as they may be potential treatment targets. In particular the authors have validated components of the mitoribosomal large subunit in the model apicomplexan Toxoplasma gondii.

Although, the thematic of the manuscript is interesting and pertinent, the manuscript needs revision before acceptance for publication.

Major issues:

Please revise all the manuscript according to the rule: When writing scientific names: italicize family, genus, species, and variety or subspecies. Begin family and genus with a capital letter. Kingdom, phylum, class, order, and suborder begin with a capital letter but are not italicized. In the manuscript there are several examples reference to a genus (“Toxoplasma”, “Plasmodium” and “Trypanosoma”) but not italicized. Please revise the manuscript, tables and figures and correct.

Example: Line 188 – “Plasmodium falciparum” should be formatted in italic

Please also notice that symbols for genes should be italicized whereas symbols for proteins are not italicized. Gene names that are written out in full are not italicized, but genotype designations should be italicized, whereas phenotype designations should not be italicized. Please revise your manuscript, tables and figures accordingly.

Please revise and correct all figure’s order of appearance in the text and captions:

Line 215 – First figure in the manuscript is named “ Figure 3” and is not referred in the text above figure. It is described in manuscript line 285-304. Please revise.

Figure´s captions should be more informative to the reader. For example in figure 3, describe in full what is “BN+PAGE + Western“. The caption refers “Complex V assembly” in the figure it appears as “ATP Syn”. Please make the connection between the concepts to improve reader’s understanding. The authors may also add the main information/conclusion draw from the image. This will help a less experienced reader in following the manuscript.

Line 247 – “figure 1” caption – Improve figure’s caption information to the reader. Figure 1A is not perceptible. What are bands? What are their molecular weight? Why there are different primers? Please revise. Figure 1B – add scale bars to the images and provide magnification information in the description. Figure 1C, please add in full in the description “DDM” and “Digi”. Figure 1D, add the information for IP-RNA. Please remove the reference to supplementary figures from the figure caption (maintain only in the text description) and add the summary of the main information/conclusion draw from the image.

Line 276 – Figure 2B, Add molecular weight information; Figure 2C- improve image quality of the parasite plates. Provide a scale bar and magnification information. Also, an arrow identifying the parasites may also be o interest as it helps a less experienced reader.

Regarding Supplementary figure 3, as it constitutes a control for the results presented in figure 3, I suggest to the authors to present it together with Figure 3, as it might improve reader’s understanding. A careful revision of figure description and information given to the reader is needed.

Section 3.1 – This section contains the detail description of the strategy followed by the authors to identify the new targets (putative mitoribosomal proteins for T. gondii). As to facilitate the reader’s understanding, I suggest that the authors present an experimental design flow of the strategy followed to illustrate the text, helping focus reader’s attention.  

The authors presented statistical analysis (figure 2 line 281; figure 4 -line 318-319) of the obtained results, but no reference to what statistical analysis was performed is given. Please revise and add statistical analysis to the methods section.

The authors should include a “Conclusion” section, highlighting their findings main conclusions. this section is important as closure section to the manuscript.

Overall, there is some fragmentation of important concept trough the manuscript. This creates as sense of confusion in the reader. Please revise the text and improve the clarity and uniformity of the information provided.

Minor issues:

Line 151: Provide a full description of “ATc”. This the first reference to the compound, but the full description is in line 266. Please revise.

Line 162-164-Please remove the text completely.

Line 177 and line 188- “wished”, may not be the most scientific form of writing. Please replace by “aimed” or “target” or a similar form.

Line 205 –“ in silico” should be formatted in italic

Author Response

Please see the attachment - please note that the same doc was submitted in respond to both reviewer comments because it includes responses to both. 
